# Las Bolitas Syndrome in *Penaeus vannamei* Hatcheries in Latin America

**DOI:** 10.3390/microorganisms12061186

**Published:** 2024-06-12

**Authors:** Pablo Intriago, Bolivar Montiel, Mauricio Valarezo, Xavier Romero, Kelly Arteaga, Nicole Cercado, Milena Burgos, Andrew P. Shinn, Alejandra Montenegro, Andrés Medina, Jennifer Gallardo

**Affiliations:** 1South Florida Farming Corporation, 13811 Old Sheridan St, Southwest Ranches, FL 33330, USA; 2South Florida Farming Laboratory, Av. Miguel Yunez, Km 14.5 via a Samborondón, Almax 3 Etapa 1- Lote 3 Bodega 2, Samborondón, Guayas, Ecuadormilenaburgos1999@outlook.es (M.B.);; 3Ficus 302 y Antonio Sanchez, Calle 11 N-O, Guayaquil, Ecuador; 4INVE (Thailand), 471 Bond Street, Bangpood, Pakkred, Nonthaburi 11120, Thailand; 5Centre for Sustainable Tropical Fisheries and Aquaculture, James Cook University, Townsville, QLD 4811, Australia

**Keywords:** las bolitas syndrome, bacterial infection, zoea, post-larvae, hepatopancreas infection

## Abstract

In September 2023, several hatcheries in Latin America experienced significant mortality rates, up to 90%, in zoea stage 2 of *Penaeus vannamei*. Observations of fresh mounts revealed structures resembling lipid droplets, similar to those seen in a condition known as “las bolitas syndrome”. Routine histopathological examinations identified detached cells and tissues in the digestive tracts of affected zoea, contrasting with the typical algal cell contents seen in healthy zoea. Polymerase chain reaction (PCR) testing for over 20 known shrimp pathogens indicated minimal differences between diseased and healthy batches. Both groups tested negative for acute hepatopancreatic necrosis disease (AHPND) but positive for *Vibrio* species and Rickettsia-like bacteria in the diseased samples. Histological analyses of the affected zoea revealed characteristic tissue degeneration in the hepatopancreas, forming spheres that eventually migrated into the upper gut, midgut, and midgut caeca, a pathology identified as bolitas syndrome (BS). Microbiological assessments revealed *Vibrio* species at concentrations of 10^6^ CFU zoea/g in affected zoea, approximately two orders of magnitude higher than in healthy zoea. Bacterial isolation from both healthy and BS-affected zoea on thiosulphate–citrate–bile salts–sucrose (TCBS) agar and CHROMagar™ (Paris, France), followed by identification using API 20E, identified six strains of *Vibrio alginolyticus*. Despite similarities to “las bolitas syndrome” in fresh mounts, distinct histopathological differences were noted, particularly the presence of sloughed cells in the intestines and variations in hepatopancreatic lobes. This study highlights the critical need for further research to fully understand the etiology and pathology of bolitas syndrome in zoea stage 2 of *P. vannamei* to develop effective mitigation strategies for hatchery operations.

## 1. Introduction

During the late 1980s, 1990s, and the early 2000s, the most predominant pathologies in hatcheries in Latin America were Larval Bolitas Syndrome (LBS), Zoea 2 Syndrome, and Mysis Molt Syndrome [1,2,3]. During the same period, the main pathologies reported in post-larvae were luminescent bacteria [4,5,6,7]. After 2015, disease outbreaks with high rates of mortality were more commonly seen in post-larval production, e.g., post-larvae AHPND (PL-AHPND), translucent post-larva disease (TPD), etc. [8,9,10,11]. LBS is a condition firstly characterised by a distinctive pathology of the hepatopancreas, where there is a cellular sloughing of the epithelium of the hepatopancreas, forming spheres, which eventually move into the upper gut [1]. Gross pathology of LBS normally develops in a matter of hours, from healthy well-fed zoea to empty moribund animals. At the same time, the larvae become bioluminescent, which is accompanied by changes in behaviour and in a loss of appetite [2]. Although LBS may occur during post-larval (PL) stages, massive mortalities, reaching up to 90%, have been observed primarily in the earlier zoea and mysis stages. A distinctive characteristic of LBS is the presence of bolitas in the hepatopancreas, which eventually migrate to the intestine.

LBS has been associated with infection of *Vibrio* spp. [2,12]. Zoea 2 syndrome was only associated with *V. harveyi* and *V. alginolyticus* [3,13], and the species associated with mysis syndrome has never been identified. The nocturnal luminescence seen in hatcheries in both Asia and the Americas was typically attributed to luminescent *Vibrio* [2,4,5,6,7]. In more recent times, outbreaks of sudden and acute mortalities in penaeid shrimp hatcheries typically start in the PL stages, from an active and apparently healthy state to moribund and dead. The speed and virulence at which these massive mortalities occur have been observed in many production facilities, and in most cases have been associated with, or linked to, different strains of *Vibrio* [8,9,10]. Intriago et al. [10] provided evidence that the cause of these rapid mortality events was a species of *Vibrio* carrying the same plasmids as the *Vp*AHPND reported as causing acute hepatopancreatic necrosis disease (AHPND) in culture ponds elsewhere. The condition was tentatively named post-larvae AHPND (PL-AHPND) to differentiate it from other pathologies affecting penaeid shrimp larvae. In Asia, translucent post-larva disease (TPD) has been the main cause of larval disease and mortality, where the causative agent is a strain of *V. parahaemolyticus* carrying a haemolysin gene (isolate Vp-JS20200428004-2) [8,9,11]. In India, a similar condition referred to as zoea 2 syndrome was described affecting zoea 2 of *Penaeus vannamei* [14].

In September 2023, some culture facilities in the Latin America region experienced high mortalities in zoea stages 2 and 3. Given the high mortality rate, these tanks were discarded. The causal agent associated with those mortalities has not been reported. Microscopic observation of the larvae revealed the presence of “bolitas” (spheres) in the hepatopancreas. Generally, the clinical indicators and the macroscopic appearance on wet mounts aligned with what was previously identified as LBS. The present study reports the microbiological, polymerase chain reaction (PCR) tests and histological findings of healthy and of diseased animals suffering from this this condition.

### Rationale of the Study

Samples were sent by clients for monitoring shrimp health status or disease outbreaks in their hatcheries. Animals (zoea) used as controls were grossly healthy and were collected as reference specimens to compare with the affected animals. This study was not an epidemiological study but a simple prevalence analysis of various shrimp pathogens in randomly provided samples from a hatchery that reported mortality. Unfortunately, owners and managers do not disclose information regarding the protocols used or the physical and chemical parameters of the culture, as they believe it could damage the hatchery’s reputation. Consequently, we are unable to conduct more extensive sampling or provide detailed information related to the larval culture.

## 2. Materials and Methods

### 2.1. Sample Collection

Samples of *P. vannamei* larvae from two hatcheries in Latin America were sampled (precise location details are withheld acknowledging the facilities request for confidentiality). Samples for microbiology, PCR and histopathology were taken from two tanks from each hatchery. Hatcheries were selected because one of the hatcheries had reported heavy mortality at zoea stages 2–3, while the other hatchery, with apparently healthy zoea 2–3, was selected as the control. It should be noted that shrimp sampled for PCR and histology were different individuals from the same populations. To protect client privacy, the country and the precise location of each culture facility from which the samples were obtained will not be disclosed.

### 2.2. Microbiology of the Larvae

The total concentration of bacteria in the larvae was determined by first bathing the larvae (i.e., 1 g or approximately >2000 zoea stage 3) held in a 200 um nylon sieve for 2–3 min in a solution of 50 ppm active chlorine prepared using 35 ppt seawater. The zoea were then rinsed with excess sterile seawater. The weight of the larvae was recorded using a Metler digital balance accurate to 0.01 g and then placed in a mortar with approximately 1 g of autoclaved beach sand and 10 mL of sterile seawater which was used to facilitate the grinding of the bulk of larvae. After grinding the sample, it was sequentially diluted in test tubes with sterile seawater to 1 × 10^2^, 10^3^ and 10^4^. Then, 100 µL of the relevant dilution was placed per duplicate on Petri dish plates containing either tryptic soy agar (TSA, Difco) or thiosulphate–citrate–bile–sucrose (TCBS, Difco) agar or on CHROMagar^TM^ *Vibrio* and then incubated for 24–48 h at 30 °C. Thereafter, the number of colony-forming units (CFU) on each plate was recorded. Dilutions were based on obtaining >20 to <200 colonies per plate. Bacterial identification was performed and biochemical characteristics were examined using an API 20E Kit [15].

### 2.3. PCR Methods

DNA was extracted from whole larvae fixed in 95% alcohol using an Omega (Omega Bio-tek, Inc., Norcross, GA, USA), Bio-Tek E.Z.N.A. tissue DNA kit following the manufacturer’s protocol. In brief, each 1 g sample was ground using a microcentrifuge pestle. Approximately 200 mg of the tissue was then transferred to a clean 1.5 mL Eppendorf tube, then 500 μL of tissue lysis buffer (TL, Omega, Bio-Tek E.Z.N.A.) and 25 μL of Omega Biotek (OB, Omega, Bio-Tek E.Z.N.A) protease solution were added. The sample was then vortexed and then incubated in a thermoblock at 55 °C for approximately 3 h with vortexing every 30 min. RNA was removed by adding 4 μL of RNase A (100 mg/mL), then mixing, then keeping the sample at room temperature for 2 min. The sample was then centrifuged at 13,500 RPM for 5 min and the supernatant carefully transferred to a new 1.5 mL Eppendorf tube. To this, 220 μL of BL buffer (Omega, Bio-Tek E.Z.N.A.) was added, and the mixture was vortexed and incubated at 70 °C for 10 min. Thereafter, 220 μL of 100% ethanol was added and vortexed, and the contents were passed through a HiBind^®^ DNA Mini Column into a 2 mL collection tube. The columns were then centrifuged at 13,500 RPM for 1 min, and the filtrate was discarded. Subsequently, 500 μL of HBC buffer (Omega, Bio-Tek E.Z.N.A), diluted with 100% isopropanol, was added to the column, and the sample was spun at 13,500 RPM for 30 s. The filtrate was discarded, the column was washed twice with 700 μL of DNA wash buffer diluted with 100% ethanol and the sample was centrifuged at 13,500 RPM for 30 s. The filtrate was discarded. This step was repeated. The column was then centrifuged at 13,500 RPM for 2 min to dry it out. The dried column was placed in a new nuclease-free 1.5 mL Eppendorf tube, and 100 μL of elution buffer, which was heated to 70 °C, was added to the column. The sample was allowed to sit for 2 min before being centrifuged at 13,500 RPM for 1 min. This elution step was repeated. The eluted DNA was then stored at −20 °C until required.

RNA was extracted from whole larvae, tissue or organs fixed in 90% alcohol following the manufacturer’s protocol (Omega, Bio-Tek E.Z.N.A. Total RNA Kit (TRK)). Approximately 200 mg of tissue was then moved to a clean 1.5 mL Eppendorf tube. To this, 700 μL TRK Lysis Buffer (Omega, Bio-Tek E.Z.N.A) was added, and the tube was left at room temperature for approximately 3 h with vortexing every 30 min. The sample was then centrifuged at 13,500 RPM for 5 min, and the supernatant was carefully transferred to a new 1.5 mL Eppendorf tube to which 420 μL of 70% ethanol was added. After vortexing to mix thoroughly, the contents were passed through a HiBind^®^ RNA Mini Column into a 2 mL collection tube. The columns were then centrifuged at 13,500 RPM for 1 min, and the filtrate was discarded. Subsequently, 500 μL of RNA Wash Buffer I (Omega, Bio-Tek E.Z.N.A) was added to the column, and the sample was spun at 13,500 RPM for 30 s. The filtrate was discarded, and the column was washed twice with 500 μL RNA Wash Buffer II and diluted with 100% ethanol. The column was then centrifuged at 13,500 RPM for 1 min to dry it out. The filtrate was discarded. This step was repeated. The column was then centrifuged at 13,500 RPM for 2 min to dry it out. The dried column was placed in a new nuclease-free 1.5 mL Eppendorf tube, and 70 μL of nuclease-free water was added to the column. The sample was centrifuged at 13,500 RPM for 2 min. This elution step was repeated. The eluted RNA was then stored at −70 °C until needed. The pathogens tested are listed below.

PCR analyses were conducted for the following pathogens in animal samples: hepanhamaparvovirus (DHPV) [16]; *Macrobrachium* bidnavirus (MrBdv) [17]; decapod iridescent virus 1 (DIV1) [18]; white spot syndrome virus (WSSV) [19]; infectious hypodermal and haematopoietic necrosis virus (IHHNV) [20,21,22,23]; Wenzhou shrimp virus 8 (WzSV8) [24]; *P. vannamei* nodavirus [25]; covert mortality nodavirus (CMNV) [26]; infectious myonecrosis virus (IMNV) [27]; yellow head virus (YHV) [28]; gill-associated virus (GAV) [28]; Taura syndrome virus (TSV) [29,30]; *Macrobrachium* nodavirus (MrNV) [31]; extra small virus (XSV) usually associated with MrNV [32]; *Spiroplasma* [33]; *Vibrio* spp. (*Vibrio*-specific 16S rRNA gene fragment) [34]; *V. parahaemolyticus* (collagenase gen) [35]; *V. parahaemolyticus* (gen vhvp-1and gen vhvp-2) [36]; *V. parahaemolyticus* (gen tdh) [37]; *V. harveyi* (gen vhh) [37]; *Rickettsia*-like bacteria (RLB) [38]; necrotizing hepatopancreatitis bacteria (NHP-B) [39]; *Ecytonucleospora* (*Enterocytozoon*) *hepatopenaei* (EHP) [40]; non-EHP Microsporidia [41]; acute hepatopancreatic necrosis disease (AHPND) [42]; and Haplosporidia [43].

Additionally, the following pathogens were screened for from isolated bacteria: *Vibrio* spp. (*Vibrio*-specific 16S rRNA gene fragment) [34]; *V. parahaemolyticus* (collagenase gen) [35]; *V. parahaemolyticus* (gen vhvp-1and gen vhvp-2) [36]; *V. parahaemolyticus* (gen tdh) [37]; *V. harveyi* (gen vhh) [37]; and toxR gen [44].

### 2.4. Histopathology

For histological analysis, samples were prepared following the procedures outlined by Bell and Lightner [45]. Briefly, larvae were fixed in Davidson’s alcohol, formalin and acetic acid (AFA) using at least 1 g of larvae from each tank. These were fixed for at least 24 h before processing for routine tissue embedding and histological sectioning. The 5 µm thick tissue sections were then stained with hematoxylin and eosin (H&E) and with Twort’s Gram stain to differentiate Gram-positive from Gram-negative bacteria (CP Lab Chemicals, Novato, CA, USA). For each sample of larvae collected from each tank, one block was prepared, with each block containing approximately 1 g of zoea stages 2–3. Four tissue sections were cut from each paraffin block.

## 3. Results

Microscopic observation of the larvae revealed the presence of “bolitas” (spheres) in the hepatopancreas (Figure 1a–d). In fresh mounts of normal healthy zoea 2–3, algal material was seen in the digestive tract, giving it a brown-gold colouration (Figure 1a). By comparison, the affected/diseased larvae had an empty digestive tract and the presence of a round-shaped material that appeared as lipid droplets (Figure 1b–d) and lack of food in the intestine. This general clinical sign is the one usually described by shrimp hatchery technicians and biologists for what has been known as “las bolitas syndrome”.

### 3.1. PCR Results

PCR analysis of both the healthy and affected samples of zoea was negative for seventeen known shrimp pathogens, including DHPV, MrBdv, SHIV, WSSV, PvNV, CMNV, IMNV, YHV, GAV, TSV, MrNV, XSV, *Spiroplasma*, NHPB, EHP, AHPND and Haplosporidia (Table 1). The apparently healthy zoea, however, were positive for IHHNV EVE (2/2), WzSV8/PvSV (2/2), RLB (1/2) and Microsporidia (1/2). By comparison, the LBS-affected zoea were positive for IHHNV EVE (1/2), for *Vibrio* spp. (2/2), for RLB (2/2) and for Microsporidia (1/2). In summary, the only difference between the two sets of samples was the detection of *Vibrio* in the zoea affected with LBS and WzSV8 in the healthy zoea.

### 3.2. Microbiology

The concentrations of total heterotrophic bacteria (TSA) and presumptive *Vibrio* (TCBS and CHROMagar^TM^ *Vibrio*) are shown in Table 2. The concentrations of total bacteria in the affected zoea were almost an order of magnitude higher than that determined for the sample of healthy zoea. Presumptive *Vibrio* values were almost two orders of magnitude higher in the affected zoea than in the healthy zoea. Presumptive *Vibrio* were found to represent 17% and 6% of the total bacteria population in the unhealthy and healthy zoea, respectively. Green colonies (on TCBS) and purple colonies (on CHROMagar^TM^ *Vibrio*) represented 0.2% and 82% of the those recovered from the affected zoea, while a reverse picture of 56% and 1% was found from the analysis of the healthy zoea (Table 2).

A total of eleven strains were isolated from the LBS-affected zoea and three from the healthy zoea (Table 3a). Biochemical profiling of the eleven strains recovered from the affected zoea identified six as *V. alginolyticus*, two as *V. fluvialis* and one as *V. vulnificus*, one *Aeromonas* (undetermined) and one *Pasteurella* (undetermined). Of the three strains isolated from the healthy zoea (Table 3a), one strain was identified as *V. alginolyticus*, one as *V. fluvialis* and one as *Aeromonas*. Interestingly, all 14 strains were positive for *Vibrio* by PCR, which suggests that 27% of the results returned by the API 20E biochemical profiles were false negatives. None of the 14 bacteria tested positive for the specific genes of *Vibrio parahaemolyticus*, including ToxR, Vhp-1, and Vhp-2, which encode the highly virulent proteins of *V. parahaemolyticus* (Table 3a). The strains identified as *Aeromonas* and *V. fluvialis* by the API 20E (LBS-9 and H-2) but as *Vibrio* by PCR were Pir AB positive (AHPND). The strain H-3 identified by API 20E as *A. hydrophila* was positively identified as *V. harveyi* by PCR.

Biochemical characterisation of all 13 isolated *Vibrio* is shown in Table 3b. This table is divided between two types of *Vibrio* isolated from LBS zoea (*V. alginolyticus* and *Vibrio* spp. identified by API 20E) and three species of *Vibrio* isolated from the healthy zoea. The availability of fermentation/oxidation of arabinose was the main difference between *Vibrio* spp. isolated from LBS zoea and the other two groups of bacteria, namely *V. alginolyticus* from LBS and *Vibrio* spp. from healthy zoea. On the other hand, *V. alginolyticus* from LBS-affected zoea can be differentiated from the other two groups in their inability to ferment.

### 3.3. Histopathology

Eight microscopy slides stained with haematoxylin and eosin (H&E) were prepared, containing approximately 120 larvae each from normal, healthy zoea stages 2–3 and diseased tanks. These slides were thoroughly examined to identify histopathological differences between healthy and diseased larvae.

Observations in Healthy Zoea 2–3 Larvae:Digestive Tract Content: Healthy zoea 2–3 larvae exhibited feed debris within their digestive tracts. The digestive system at this stage is still developing, and the presence of feed debris indicates normal feeding and digestive processes.Hepatopancreas Development: Early development of the hepatopancreas was evident through the presence of lateral lobes, which are precursors to the fully functional hepatopancreas [47]. These lobes were clearly visible in the tissue sections (Figure 2a,b).Peritrophic Membrane Integrity: An intact peritrophic membrane was observed in healthy larvae. This membrane plays a crucial role in protecting the digestive epithelium and facilitating digestion.Algal Material: The algal material consumed by the larvae at this stage was seen as small-sized brown debris within the digestive tract (Figure 2c,d). This indicates that the larvae were actively feeding on algal matter, which is typical for this developmental stage.Absence of Detached Cells: No detached cell material was observed in the lumen of the developing digestive system of healthy larvae. The absence of detached cells indicates a lack of tissue degeneration or pathology within the digestive tract (Figure 2c,d).

Epithelial Cell Detachment and Necrosis: Most LBS-affected zoea showed detachment of epithelial cells within the lumen and structural loss due to necrosis in the hepatopancreas (Figure 3a–c). This detachment ranged from minimal to severe and contrasted sharply with the intact epithelia in healthy larvae (Figure 3d–f).Midgut Epithelium: In LBS-affected larvae, there was notable detachment of midgut epithelial cells into the lumen. This pathological feature was absent in healthy larvae, where the midgut epithelium remained intact.Peritrophic Membrane: Cells were observed inside the peritrophic membrane in LBS-affected larvae (Figure 3g). This is indicative of disruption in the peritrophic membrane’s protective role.Gram-Negative Bacteria: Twort’s stain revealed the presence of Gram-negative bacteria, with or without sloughed material, in the affected larvae (Figure 4a,b). This finding was consistent with microbiological analyses showing higher concentrations of *Vibrio* spp. in diseased zoea.Intestinal Differences: Significant differences were noted in the intestine area between normal and LBS-affected larvae. In normal larvae, the intestines appeared healthy with no bacterial overgrowth and intact peritrophic membranes (Figure 2c). In contrast, LBS-affected larvae had an abundance of Gram-negative bacteria and compromised peritrophic membranes (Figure 4d).Absence of Bolitas: During this study, no bolitas (i.e., spheres) were observed in any histological slides, and such structures have not been documented in the literature [1,2]. It is suggested that the bolitas may be lipidic in nature, given that tissue sample preparation for histology involves processing through several solvents that could potentially dissolve lipid-based structures.

## 4. Discussion

The present study describes the histopathological lesions, PCR and microbiology of samples of *P. vannamei* zoea collected from two different hatcheries, one with an outbreak of LBS and the other with apparently healthy animals. Of all pathogens that were analysed by PCR, the only key differences between the LBS and healthy zoea were the detection of *Vibrio* and more abundant RLB in the diseased zoea. The presence of the WzSV8 virus in the healthy zoea requires more study; this RNA virus has been found in a wide range of different environments and regions including wild broodstock, and its effect in penaeid production needs to be the subject of further studies [48].

Microbiology showed that the total number of culturable bacteria (TSA) was one order of magnitude higher in the affected zoea and the number of presumptive *Vibrio* was almost two orders of magnitude higher when compared to the healthy zoea. Presumptive *Vibrio* represented 17% and 6% of the total bacterial population in the unhealthy and healthy zoea, respectively. Green colonies (on TCBS) and mauve colonies which are presumptive *V. parahaemolyticus* (on CHROMagar^TM^ *Vibrio*) represented 0.2% and 82% of the total *Vibrio* count in affected zoea, while a reverse pattern was seen in the healthy zoea which had 56% and 2%, respectively (Table 2). This finding is interesting because the common assumption is that green colonies on TCBS and purple colonies on CHROMagar^TM^ *Vibrio* typically represent *V. parahaemolyticus*. In this regard, Soto-Rodriguez et al. [49], studying the phenotypic characteristics and growth kinetics of three *V. parahaemolyticus* strains with different virulence and one non-pathogenic strain, found that independent of the virulence of the strain, a high metabolic diversity was present, which yielded different-coloured phenotypes on the CHROMagar™ *Vibrio*.

It is important to note that all 14 strains were identified by PCR as members of the genus *Vibrio*, so it can be concluded that API 20E results (21% false identification at a genus level) were not reliable and should be taken with caution (Table 3a). While Overman et al. [50] stated that the API 20E is a valid system for use in the identification of the more commonly occurring members of the family Vibrionaceae, the system has been reported to result in false negatives [51,52]. API identification is based on biochemical profiles, but it has been found that biochemical profiles and genotype are not necessarily associated with virulence potential [53]. The relationship between the variation or differences in the API 20E identifications and the possible virulence and genus variation of each strain is something that requires further exploration.

Table 3b provides details regarding the characterisation of the isolated bacteria using the API 20E system. This table is divided between in two groups: bacteria isolated in LBS zoea (*V. alginolyticus* and *Vibrio* spp. with six and five strains, respectively) and bacteria (three strains) isolated from healthy zoea. Fermentation/oxidation of arabinose and amygdalin was different between the three groups of *Vibrio* isolated from affected vs. healthy zoea (Table 3b). Acetoin production was the main difference between *V. alginolyticus* and other species of *Vibrio* isolated from healthy and affected zoea. The ability of acetoin production by *Vibrio* spp. isolated from LBS-affected animals is interesting, as generally this metabolite is produced when microorganisms employ the 2,3-butanediol pathway to ferment sugars, and this pathway generates less acidic and more neutral end products, such as acetoin and 2,3-butanediol. As acetoin is a neutral fermentation product and this biosynthetic reaction consumes intracellular protons, bacterial growth can occur on a glucose carbon source without pH decrease [54,55]. In addition, inhibition of acetoin production has been suggested as a potential mechanism to control the pathogenic *V. cholerae* that is known to be acid sensitive [55]. Differentiation of *Vibrio* strains by their biochemical properties is not rare. For instance, *V. anguillarum* could be separated mainly based on its reaction on indole production and the fermentation of amygdalin and arabinose [56].

A comparison between the histology in this study and the first report of LBS described by Morales [1] is impossible. Unfortunately, only transmission electron microscopy (TEM) images were presented, and no tissue sections stained with H&E were recorded. The publication by Robertson et al. [2] shows the presence of melanised necrotic bundles in the hepatopancreatic tubules (Figure 2 in [2]). This feature was never seen during the present research. The most likely reason is that the animals used for the histology were not zoea but were least at PL1 stage when comparing the anatomy of those that were presented to other detailed descriptions regarding the larval development of *P. vannamei* [57]. Important differences between the pathology presented here for LBS and for PL-AHPND [10] is that the there is no massive sloughing of cells in the hepatopancreas as was described for PL-AHPND. In addition, PL-AHPND was never found or described in zoea [10]. In addition, no PirAB was detected nor other pathogenic genes. A similar material present in shrimp larvae reared in China and suffering mortalities during zoea stage 2 has been described by other authors; in their case this was associated with a strain of *V. alginolyticus* [58].

Historically, the Latin American shrimp sector has followed the Ecuadorian model, where post-larvae are produced from broodstock obtained from production ponds. Broodstock are selected based on weight and then transferred to a hatchery to produce the next generation of post-larvae. This process gives little to no long-term control over biosecurity [59]. In addition, high concentrations of *Vibrio* spp. can be very common in non-biosecure penaeid hatcheries or those that are not specific pathogen free, both free and as attached to the larvae. From hatching through to harvest, the microbiological environment is a soup of bacteria and virus-like particles (VLPs) [60,61,62]. As the larvae transition from a diet based on algae as zoea to animals that source proteins as mysis, the larvae then undergo a dramatic change in the volume of the hepatopancreas and the biochemistry of the digestive enzymes [63]. In zoea, the filtration of particles is almost indiscriminate. As the zoea stages are exposed to very high concentrations of bacteria, free-living and attached, including *Vibrio* that are easily ingested and able to pass into the digestive tract [64], the detrimental effects of such can be observed depending on several factors such as the larval sub-stage involved, the *Vibrio* species present and their concentration [65].

Intriago and Jimenez [66] replicated the bolitas syndrome in *P. vannamei* zoea using a luminescent strain of *V. harveyi* at concentrations as low as 10^3^ cell/mL. Interestingly, this strain was isolated from diseased farm-cultured *P. vannamei* affected by haemocytic enteritis. They postulated that pathogens could be bouncing back from hatcheries to the broodstock or vice versa and that the differences found in the histopathology between the larvae and adults could have been attributed on one hand due to the differences in the degree of organ development and on the other hand to the pathogen species, its virulence and its concentration.

The key event in the appearance of diseases could be attributed to stress (temperature, salinity, density, toxins, etc.) because of alterations in the environment, and this exerts a change in the host–pathogen interaction and the transmission of bacteria between species. Such modifications act on pathogens to facilitate their increased transmission between individual hosts and increased contact with new host populations or species and on the selection, pressure leading to the dominance of pathogen strains adapted to these new environmental conditions [46,67]. Unfortunately, there is no way to compare the histology of this study with that of previous published reports, although the histology of this study resembles the zoea 2 syndrome reported by Kumar et al. [14]. However, the differences in pathology could be the product of different responses by the host to a wide dynamic bacteria genotype and the concentration of bacterial pathogens [12]. In addition, two pathogens could also produce the same macroscopic pathology (LBS) but the differences in the damage at the tissue level will depend on all the factors described above.

## 5. Conclusions

From the above information is clear that no virus was involved in this disease. Although some were detected by PCR, no inclusion bodies or associated lesions were found. We can also eliminate common *Vibrio* pathologies such AHPND or the new highly lethal *V. parahaemolyticus* described for *P. vannamei* larvae carrying the genes Vhp1- and Vhp-2. All 14 bacterial strains isolated were identified as *Vibrio* by PCR. It is tempting to suggest that Vibrio played a role in the pathogenicity, however, we cannot rule out RLB or toxicity of the water. We can, however, conclude that *Vibrio* species as pathogenic or as opportunistic bacteria play an important role in both LBS and healthy tanks, and in general they were characterised by high metabolic diversity which yielded different-coloured phenotypes on the CHROMagar™ *Vibrio*. The histopathological examination of LBS-affected zoea stages 2–3 larvae revealed significant pathological changes, including epithelial cell detachment, hepatopancreatic necrosis, disrupted peritrophic membranes and the presence of Gram-negative bacteria. These findings highlight the severe impact of LBS on the digestive system of *Penaeus vannamei* larvae, contrasting starkly with the intact structures and absence of pathogenic features in healthy larvae. Further research is needed to fully understand the etiology of LBS and develop effective interventions to mitigate its effects in shrimp hatcheries.

## Figures and Tables

**Figure 1 microorganisms-12-01186-f001:**
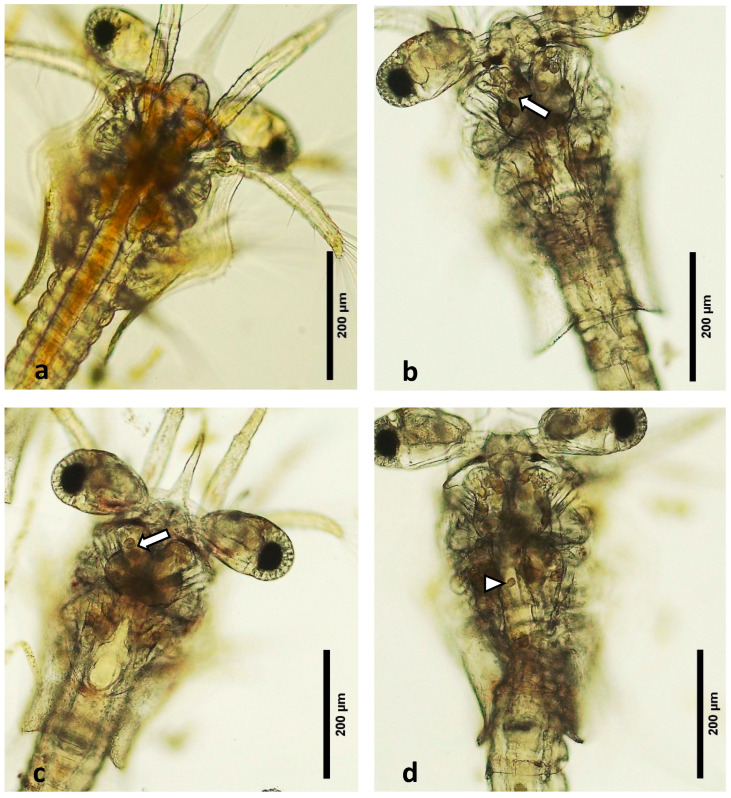
Fresh mount of zoea 2 larvae. (**a**) A healthy larva with a normal digestive tract. (**b**–**d**) Affected larva, note the lack of food contents and the presence of “bolitas” (arrows) in hepatopancreas, as well as upper gut (arrowhead).

**Figure 2 microorganisms-12-01186-f002:**
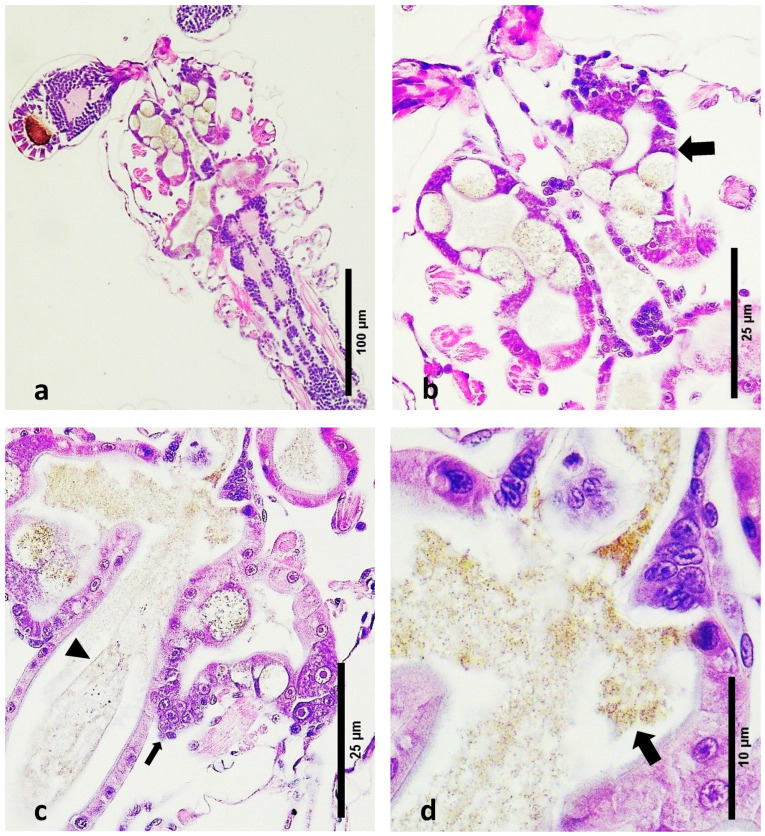
Haematoxylin- and eosin-stained sections through normal and affected zoea 2 larvae. (**a**) Normal zoea 2 larva. (**b**) Closer look at image a, showing the early formation of the hepatopancreas (thick arrow). Note the food content of the digestive system and the absence of tissue debris. (**c**) Lobes that will form the hepatopancreas (thin arrow). Note the intact peritrophic membrane and the food content (arrowhead). (**d**) Closer view of the digestive tract with brown particles of possible microalgae (thick arrow).

**Figure 3 microorganisms-12-01186-f003:**
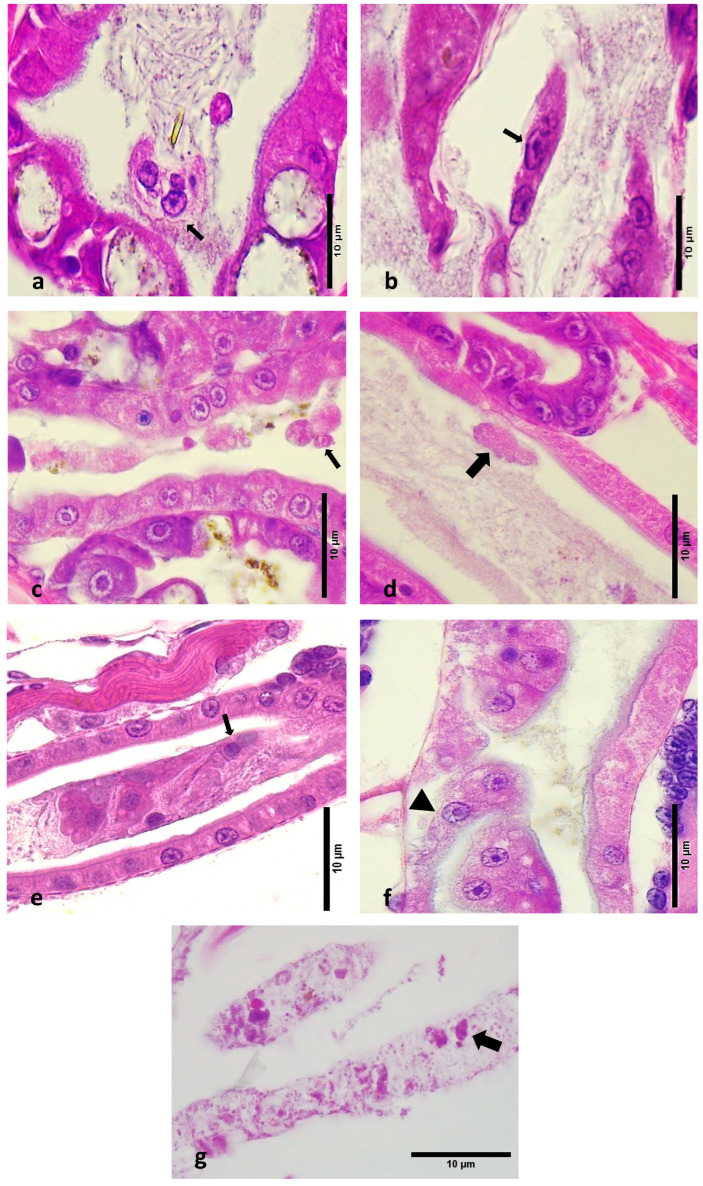
Haematoxylin- and eosin-stained sections of normal and affected zoea 2 larvae. (**a**) Hepatopancreas of affected zoea 2–3 larvae. Note the detachment of cells within the lumen (thin arrow). (**b**) Structural loss due to necrosis in the hepatopancreas (thin arrow). (**c**) Hepatopancreas and intestinal track area, presence of sloughed material (thin arrow). (**d**) Sloughing of midgut epithelial cells into the lumen, note the outside of the peritrophic membrane (thick arrow). (**e**,**f**) Intestine area with severe sloughed material (thin arrow), including basement membrane (arrowhead). (**g**) Faecal strands with sloughed cellular material (thick arrow).

**Figure 4 microorganisms-12-01186-f004:**
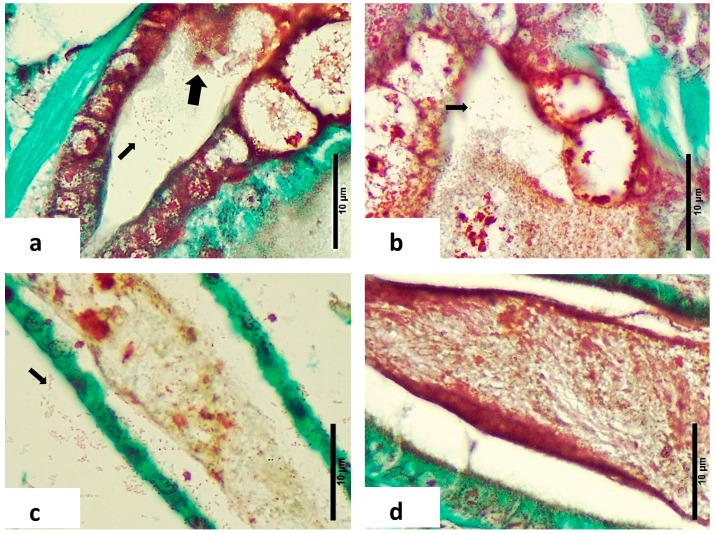
Twort’s Gram-stained sections through normal and affected zoea 2 larvae. (**a**) Affected zoea hepatopancreas with sloughed material (thick arrow) and bacterial cells in lumen (thin arrow), note the red colour which indicates Gram-negative bacteria. (**b**) Normal zoea hepatopancreas with bacterial cells in lumen (thick arrow). (**c**) Affected zoea, intestine area with an abundant presence of bacteria (thin arrow). (**d**) Intact peritrophic membrane of normal zoea. Note the absence of bacteria.

**Table 1 microorganisms-12-01186-t001:** PCR results of DNA, RNA, bacteria, EHP and non-EHP Microsporidia and Haplosporidia in 1 g of zoea from both LBS and healthy tanks.

DNA Virus													
	HPV ^1^	MrBdv ^2^	DIV1 ^3^	WSSV ^4^	IHHNV ^5^				Virus *^6^	EVE **^6^			
					309 FR	392 F/R	389 F/R	77012/773					
LBS	0/2 (0%)	0/2 (0%)	0/2 (0%)	0/2 (0%)	1/2 (50%)	1/2 (50%)	1/2 (50%)	0/2 (0%)	0/2 (0%)	1/2 (50%)			
Healthy	0/2 (0%)	0/2 (0%)	0/2 (0%)	0/2 (0%)	2/2 (100%)	2/2 (100%)	1/2 (50%)	0/2 (0%)	0/2 (0%)	2/2 (100%)			
**RNA Virus**													
	**WzSV8 ^7^**	**PvNV ^8^**	**CMNV ^9^**	**IMNV ^10^**	**YHV ^11^**	**GAV ^12^**	**TSV ^13^**	**MrNV ^14^**	**XSV ^15^**				
LBS	0/2 (0%)	0/2 (0%)	0/2 (0%)	0/2 (0%)	0/2 (0%)	0/2 (0%)	0/2 (0%)	0/2 (0%)	0/2 (0%)				
Healthy	2/2 (100%)	0/2 (0%)	0/2 (0%)	0/2 (0%)	0/2 (0%)	0/2 (0%)	0/2 (0%)	0/2 (0%)	0/2 (0%)				
**Bacteria/Microsporidia/Haplosporidia**													
	**Spirop ^16^**	**Vibrio ^17^**	**Vp ^18^**	**Vp(tdh) ^19^**	**Vp(vhp 1) ^20^**	**Vp(vhp 2) ^20^**	**Vh (vhh) ^21^**	**AHPND ^22^**	**RLB ^23^**	**NHP ^24^**	**EHP ^25^**	**Microsp ^26^**	**Haplosp ^27^**
LBS	0/2 (0%)	2/2 (100%)	0/2 (0%)	0/2 (0%)	0/2 (0%)	0/2 (0%)	0/2 (0%)	0/2 (0%)	2/2 (100%)	0/2 (0%)	0/2 (0%)	1/2 (50%)	0/2 (0%)
Healthy	0/2 (0%)	0/2 (0%)	0/2 (0%)	0/2 (0%)	0/2 (0%)	0/2 (0%)	0/2 (0%)	0/2 (0%)	1/2 (50%)	0/2 (0%)	0/2 (0%)	1/2 (50%)	0/2 (0%)

^1^ Hepanhamaparvovirus (DHPV) [16]; ^2^ *Macrobrachium* Bidnavirus (MrBdv) [17]; ^3^ Decapod Iridescent Virus 1 (DIV1) [18]; ^4^ White Spot Syndrome Virus (WSSV) [19]; ^5^ Infectious Hypodermal and Hematopoietic Necrosis Virus (IHHNV) [20,21,22,23]; ^6^ IHHNV * as infectious or ** EVE; ^7^ Wenzhou shrimp virus 8 (WzSV8) [24]; ^8^ *P. vannamei* nodavirus [25]; ^9^ Covert Mortality Nodavirus (CMNV) [26]; ^10^ Infectious Myonecrosis Virus (IMNV) [27]; ^11^ Yellow Head Virus (YHV) [28]; ^12^ Gill-associated virus (GAV) [28]; ^13^ Taura Syndrome Virus (TSV) [29,30]; ^14^ *Macrobrachium* Nodavirus (MrNV) [31]; ^15^ Extra small virus (XSV) usually associated with MrNV [32]; ^16^ *Spiroplasma* [33]; ^17^ *Vibrio* spp. *Vibrio*-specific 16S rRNA gene fragment [34]; ^18^ *V. parahaemolyticus*. Collagenase gen [35]; ^19^ *V. parahaemolyticus*. Genes vhvp-1 and vhvp-2 [36]; ^20^ *V. parahaemolyticus*. Gen tdh [37]; ^21^ *V. harveyi*. Gen vhh [37]; ^22^ Acute Hepatopancreatic Necrosis Disease (AHPND) [42]; ^23^ *Rickettsia*-Like Bacteria (RLB) [38]; ^24^ Necrotizing Hepatopancreatitis Bacteria (NHP-B) [39]; ^25^ *Ecytonucleospora* (*Enterocytozoon*) *hepatopenaei* (EHP) [40]; ^26^ non-EHP Microsporidia [41]; ^27^ Haplosporidia [43].

**Table 2 microorganisms-12-01186-t002:** Microbiology results of affected and healthy zoea.

		CFU/gr Larvae				
		TSA ^1^	TCBS ^2^	Green ^3^	CHROMagar ^4^	Purple ^5^
LBS	Tank 7	1.20 × 10^7^	4.40 × 10^5^	1.00 × 10^4^	4.80 × 10^5^	2.40 × 10^5^
	Tank 19	1.36 × 10^7^	4.00 × 10^6^	0.00 × 10^0^	3.90 × 10^6^	3.40 × 10^6^
	Average	1.28 × 10^7^	2.22 × 10^6^	5.00 × 10^3^	2.19 × 10^6^	1.82 × 10^6^
	std dev	1.13 × 10^6^	2.52 × 10^6^	7.07 × 10^3^	2.42 × 10^6^	2.23 × 10^6^
			17.3% ^6^	0.2% ^7^		83.1% ^8^
Healthy	Room 2	1.46 × 10^6^	7.60 × 10^4^	7.60 × 10^4^	6.20 × 10^4^	2.00 × 10^3^
	Room 3	8.90 × 10^5^	6.10 × 10^4^	0.00 × 10^0^	5.00 × 10^4^	1.00 × 10^0^
	Average	1.18 × 10^6^	6.85 × 10^4^	3.80 × 10^4^	5.60 × 10^4^	1.00 × 10^3^
	std dev	4.03 × 10^5^	1.06 × 10^4^	5.37 × 10^4^	8.49 × 10^3^	1.41 × 10^3^
			5.8% ^6^	55.5% ^7^		1.8% ^8^

^1^ Tryptic Soy Agar (TSA, Difco); ^2^ thiosulphate–citrate–bile–sucrose agar (TCBS, Difco); ^3^ Green colonies in TCBS; ^4^ CHROMagar^TM^ *Vibrio*; ^5^ Purple colonies in CHROMagar^TM^ *Vibrio*. ^6^ Percentage of presumptive *Vibrio* (TCBS) of total bacterial population; ^7^ Percentage of green colonies of the total population of presumptive *Vibrio* (TCBS); ^8^ Percentage of purple colonies of the total population of presumptive *Vibrio* CHROMagar^TM^ *Vibrio*.

**Table 3 microorganisms-12-01186-t003:** (a). Characterisation of bacteria isolated from affected and healthy zoea. (b). API 20E characterisation of bacteria isolated from affected and healthy zoea.

(a)													
Strain	Date	AHPND ^1^	toxR ^2^	*Vibrio* ^3^	*Vibrio* ^4^	Vhp-1 ^5^	Vhp-2 ^5^	V.p ^6^	V.hh ^7^	TCBS	CHROMagar^TM^	API 20NE Code	Identified as
LBS-8	18 September 2023	-	-	+	+	-	-	-	-	Yellow	White	4047124	*V. alginolyticus* 97.8%
LBS-1	18 September 2023	-	-	+	+	-	-	-	-	Yellow	White	4047124	*V. alginolyticus* 97.8%
LBS-10	18 September 2023	-	-	+	+	-	-	-	-	Yellow	White	4047124	*V. alginolyticus* 97.8%
LBS-11	18 September 2023	-	-	+	+	-	-	-	-	Yellow	White	4146124	*V. alginolyticus* 96.5%
LBS-6	18 September 2023	-	-	+	+	-	-	-	-	Yellow	White	4147124	*V. alginolyticus* 85.9%
LBS-5	18 September 2023	-	-	+	+	-	-	-	-	Yellow	White	4045120	*V. alginolyticus* 80.2%
LBS-2	18 September 2023	-	-	+	+	-	-	-	-	Yellow	Clear	1040127	*V. fluvialis* 76.9%
LBS-12	18 September 2023	-	-	+	+	-	-	-	-	Yellow	Purple	0040027	*V. fluvialis* 63.9%
LBS-4	18 September 2023	-	-	-	+	-	-	-	-	Green	Torquise	5046005	*V. vulnificus* 99.9%
LBS-9	18 September 2023	+	-	+	+	-	-	-	-	Yellow	White	0044427	*Aeromonas hydrophila/caviae/sobria* 1 71.1%
LBS-7	18 September 2023	-	-	+	+	-	-	-	-	Yellow	Purple	0044526	*Pasteurella multocida* 2 98.4%
H-1	23 September 2023	-	-	+	+	-	-	-	-	Yellow	White	4147125	*V. alginolyticus* 92.1%
H-2	23 September 2023	+	-	+	+	-	-	-	-	Yellow	White	1042025	*V. fluvialis* 84.1%
H-3	23 September 2023	-	-	+	+	-	-	-	+	Yellow	White	5044165	*Aeromonas hydrophila/caviae/sobria* 1 90.6%
**(b)**													
							**LBS zoea**					**Healthy**	
**API 20 E**							** *V. alginolyticus* **			***Vibrio* spp** **.**		***Vibrio* spp** **.**	
							**(6/11) ^a^**			**(5/11) ^a^**		**(3/3) ^a^**	
ONPG		ONPG		beta-galactosidase			0%			40%		67%	
ADH		arginine		arginine dihydrolase			0%			0%		0%	
LDC		lysine		lysine decarboxylase			100%			20%		67%	
ODC		ornithine		ornithine decarboxylase yellow			33%			0%		33%	
CIT		citrate		citrate utilization			0%			0%		0%	
H2S		Na thiosulfate		H2S production			0%			0%		0%	
URE		urea		urea hydrolysis			0%			0%		0%	
TDA		tryptophan		deaminase			0%			0%		0%	
IND		tryptophan		indole production			100%			100%		100%	
VP		Na pyruvate		acetoin production			100%			0%		33%	
GEL		charcoal gelatin		gelatinase			67%			20%		67%	
GLU		glucose		fermentation/oxidation			100%			60%		67%	
MAN		mannitol		fermentation/oxidation			100%			60%		67%	
INO		inositol		fermentation/oxidation			0%			0%		0%	
SOR		sorbitol		fermentation/oxidation			0%			0%		0%	
RHA		rhamnose		fermentation/oxidation			0%			0%		0%	
SAC		sucrose		fermentation/oxidation			100%			80%		100%	
MEL		melibiose		fermentation/oxidation			0%			0%		0%	
AMY		amygdalin		fermentation/oxidation			0%			80%		100%	
ARA		arabinose		fermentation/oxidation			0%			80%		0%	
		oxidase		oxidase			83%			100%		100%	
		Catalase		Catalase			100%			100%		100%	

Strains identified as LBS zoea were isolated from LBS affected zoea, and Healthy from apparently healthy zoea, or non-affected zoea. ^1^ Acute Hepatopancreatic Necrosis Disease (AHPND) [42]; ^2^ toxR gen [44]; ^3^ *Vibrio* spp. [46]; ^4^ *Vibrio*-specific 16S rRNA gene fragment [34]; ^5^ *Vibrio* highly virulent protein gen [36]; ^6^ *V. parahaemolyticus* [35]; ^7^ *V. harveyi* (vhh) [37]. - as negative + as Positive. ^a^ Positive strains/total strains.

## Data Availability

The raw data supporting the conclusions of this article will be made available by the authors on request.

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
