# Peer review of "Las Bolitas Syndrome in Penaeus vannamei Hatcheries in Latin America"

_microorganisms, 2024, doi:10.3390/microorganisms12061186_

Round 1
Reviewer 1 Report
Comments and Suggestions for Authors
This study reports the microbiological, polymerase chain reaction (PCR) tests and histological findings of healthy and of diseased animals suffering from this this condition. The control group used in the experiment consists of healthy Zoea carrying multiple viruses. Will this affect the analysis of the experiment? The iThenticate report shows a repeat rate of up to 63%, which is not original. So I can not recommend the present manuscript for publication.
Some concerns are described below:
1. Line 103-104: “Once the sample had been ground, the volume and weight were recorded again” What is the significance of this step?
2. 108: “30oC” should be revise with “30℃”
3. Line 135: “2 min” should be revised with “2 mins”.
4. Line 165: “Vibrio spp. (Vibrio specific 16S rRNA gene fragment) [37]” The reference order is incorrect.
5. Line 181-183: “Additional sections, however, were stained with three different stains: Twort’s Gram stain to differentiate Gram positive from Gram negative bacteria.” The presence of bacteria is difficult to observe in tissue pathology slides with Gram staining unless there is a large quantity of bacteria present.
6. Line 200-202: “IHHNV EVE (2/2), WzSV8/PvSV (2/2), RLB (1/2) and Microsporidia (1/2). By comparison, the LBS affected zoea were positive for IHHNV EVE (1/2), Vibrio spp. (2/2), RLB (2/2) and Microsporidia (1/2).” "What does this 1 and 2 mean? Can healthy shrimp be used as controls with so many viruses? Are they truly healthy, or just in the pre-symptomatic phase?"
7. Line 310-311: “Note the abundance Gram-negative cells and peritrophic membrane in affected vs normal zoea.” "Gram-negative cells" refers to what, and it is not discernible in the figure.
8. Figure 1,2: “Scale bar” should be indicated on the figure.
Comments on the Quality of English LanguageThe English level should be improved.
Author Response
REVIEWER 1 Please find attached the corrected manuscript
The control group used in the experiment consists of healthy Zoea carrying multiple viruses. Will this affect the analysis of the experiment?
With some exceptions, shrimp farming in Latin America is done with animals that are not SPF (specific pathogen free), and rather the industry has sought animals that, through multiple infections, acquire resistance or tolerances. Please check my last paper (Intriago, P; Medina, A; Cercado, N ; Arteaga, K; Montenegro, A; Burgos, M; Brock, J.A; Flegel, T, 2024. Passive surveillance for shrimp pathogens. 36: 102092. Doi.org/10.1016/j.aqrep.2024.102092).
The iThenticate report shows a repeat rate of up to 63%, which is not original. So I can not recommend the present manuscript for publication.
Could you please elaborate on the iThenticate report? We take this very seriously. We have just submitted to another Journal a study “Advanced Pathogen Monitoring in Penaeus vannamei from Three Latin American Regions: Passive Surveillance Part 2, which is the second part of a NEW work published this same year Intriago, P; Medina, A; Cercado, N ; Arteaga, K; Montenegro, A; Burgos, M; Brock, J.A; Flegel, T, 2024. Passive surveillance for shrimp pathogens. 36: 102092. Doi.org/10.1016/j.aqrep.2024.102092.”. we also published this novel work last year Intriago P, Medina A, Espinoza J, Enriquez X, Arteaga K, Aranguren LF, Shinn AP, Romero presence of Vibrio sp. Carrying the VpPirAB toxin genes. Aquacult Int 31: 3363-338210.1007/s10499-023-01129-0.It turns out that the coincidences were we used the same materials and methods??. In addition, we are always looking for presence of more than 20 pathogens (they are always named in all our studies). I wonder if these coincidences could be due to these new works that effectively use the same materials and methods.
Some concerns are described below:
- Line 103-104: “Once the sample had been ground, the volume and weight were recorded again” What is the significance of this step? Corrected
- 108: “30oC” should be revise with “30℃” Corrected
- Line 135: “2 min” should be revised with “2 mins”. Corrected
- Line 165: “Vibrio spp. (Vibrio specific 16S rRNA gene fragment) [37]” The reference order is incorrect. Corrected
- Line 181-183: “Additional sections, however, were stained with three different stains: Twort’s Gram stain to differentiate Gram positive from Gram negative bacteria.” The presence of bacteria is difficult to observe in tissue pathology slides with Gram staining unless there is a large quantity of bacteria present. Corrected
- Line 200-202: “IHHNV EVE (2/2), WzSV8/PvSV (2/2), RLB (1/2) and Microsporidia (1/2). By comparison, the LBS affected zoea were positive for IHHNV EVE (1/2), Vibrio spp. (2/2), RLB (2/2) and Microsporidia (1/2).” “What does this 1 and 2 mean? Can healthy shrimp be used as controls with so many viruses? Are they truly healthy, or just in the pre-symptomatic phase?” We sampled 2 samples of disease and healthy zoeas populations. 0/2 = 0%, ½ =50% and 2/2= 100%.
- Line 310-311: “Note the abundance Gram-negative cells and peritrophic membrane in affected vs normal zoea.” “Gram-negative cells” refers to what, and it is not discernible in the figure.
- Figure 1,2: “Scale bar” should be indicated on the figure. Corrected
Reviewer 2 Report
Comments and Suggestions for Authors
An interesting paper on an emerging problem in shrimp, however, it needs some additional information.
Methods
There is no mention of controls used either in the extraction of DNA and RNA or in the diseases tested for. The presence or absence of positive and negative controls should be described, especially when the tests are negative.
Histopathology Line 182-3 “additional sections were stained with three different stains: Wort's Gram stain” – but there are no other stains mentioned. What were the other two?
The “histology” would benefit by input from a histopathologist. Brown gold colouration on H&E can be a range of materials, from stain drop-out from degraded Haematoxylin through to melanin or other inert substances – including algal debris. A melanin stain would help.
In H&E staining, lipid presents as a transparent vacuole – the lipid having been extracted by the alcohol and xylene. The round-shaped material is protein or cytoplasm of some sort.
Table 1. Please add that the two samples were each of 1 gram of tissue – not one zoeal. The authors have done that for Table 2 (CPU/gr larvae).
Figure 2. There are no arrows – neither thick nor thin.
Figure 3 is problematic. As stated before, the ovoid particles staining bright pink are a proteinaceous material. Lipid presents as a vacuole, which can be detected by stains such as osmium before the alcohol/xylene steps.
The “brown-green material” has been mentioned above. It is not lipid.
The gut wall is one cell thick. it has not sloughed off (there would in that case be a hole) but there is evidence of cell death, so it could well be that death of the zoea follows an unobserved rupture of the gut.
Author Response
Please find attached the paper with corrections
Methods
There is no mention of controls used either in the extraction of DNA and RNA or in the diseases tested for. The presence or absence of positive and negative controls should be described, especially when the tests are negative.
Regarding negative controls, these are done with blanks, a blank is always run that is expected to always be negative. Regarding positive controls, by not knowing which pathogens are present, we run multiple pathogens. In theory, once the presence of a pathogen has been determined, there are some methods that would corroborate its presence, such as in situ hybridization (which we are developing for some pathogens). Now the way to generate positive controls is to sequence the amplification fragment and compare the sequences with what is expected. We also continually send samples to accredited laboratories such as the University of Arizona. We run most of the molecular analyzes with histological analyzes to corroborate that the pathogen is causing damage. Now we are an independent research laboratory without external funds, and we are limited to www.southfloridafarming.com but what we lack in funds we have in sampling coverage of commercial systems.
Histopathology Line 182-3 “additional sections were stained with three different stains: Wort's Gram stain” – but there are no other stains mentioned. What were the other two? These were fixed for at least 24 h before processing for routine tissue embedding and histological sectioning. The 5 µm thick tissue sections were then stained with hematoxylin and eosin (H&E), and with Twort’s Gram stain to differentiate Gram positive from Gram negative bacteria (CP Lab Chemicals, Novato, California USA). Corrected
The “histology” would benefit by input from a histopathologist. Brown gold colouration on H&E can be a range of materials, from stain drop-out from degraded Haematoxylin through to melanin or other inert substances – including algal debris. A melanin stain would help. Agree we have modified the description. Corrected
In H&E staining, lipid presents as a transparent vacuole – the lipid having been extracted by the alcohol and xylene. The round-shaped material is protein or cytoplasm of some sort. Agree we have modified the description. Corrected
Table 1. Please add that the two samples were each of 1 gram of tissue – not one zoeal. The authors have done that for Table 2 (CPU/gr larvae). Corrected
Figure 2. There are no arrows – neither thick nor thin. Corrected
Figure 3 is problematic. As stated before, the ovoid particles staining bright pink are a proteinaceous material. Lipid presents as a vacuole, which can be detected by stains such as osmium before the alcohol/xylene steps.
The “brown-green material” has been mentioned above. It is not lipid.
The gut wall is one cell thick. it has not sloughed off (there would in that case be a hole) but there is evidence of cell death, so it could well be that death of the zoea follows an unobserved rupture of the gut. Corrected
Round 2
Reviewer 1 Report
Comments and Suggestions for Authors
The manuscript has been revised for acceptance.